# Effect of Implantoplasty on Roughness, Fatigue and Corrosion Behavior of Narrow Diameter Dental Implants

**DOI:** 10.3390/jfb14020061

**Published:** 2023-01-21

**Authors:** Octavi Camps-Font, Jorge Toledano-Serrabona, Ana Juiz-Camps, Javier Gil, Maria Angeles Sánchez-Garcés, Rui Figueiredo, Cosme Gay-Escoda, Eduard Valmaseda-Castellón

**Affiliations:** 1Department of Oral Surgery and Implantology, Faculty of Medicine and Health Sciences, University of Barcelona, Campus de Bellvitg, C/Feixa Llarga, s/n, Pavelló Govern, L’Hospitalet de Llobregat, 08907 Barcelona, Spain; 2Bellvitge Biomedical Research Institute, Oral Surgery and Implantology, (IDIBELL), 08907 Barcelona, Spain; 3Bioengineering Institute of Technology, Facultad de Medicina y Ciencia de la Salud, Universitat Internacional de Catalunya, Sant Cugat del Vallés, 08907 Barcelona, Spain

**Keywords:** implantoplasty, fatigue, dental implant, Ti6Al4V, corrosion

## Abstract

Implantoplasty (IP) is used in dental implants with peri-implantitis and aims to remove threads and polish rough surfaces in order to prevent bacterial colonization. As a result of this procedure, implant strength might be compromised. We tested 20 tapered screw-shaped Ti6Al4V dental implants with a simulated bone loss of 50%. Ten implants underwent IP and 10 served as controls. Surface topography (S_a_, S_z_, S_sk,_ and S_dr_) was analyzed with a confocal optical microscope. Subsequently, a minimum of four series of cyclic loads were applied with a servo-hydraulic mechanical testing machine (5 × 10^6^ cycles at 15 Hz, between a maximal nominal value–starting at 529 N in the IP group and 735 N in the control group–and 10% of that force). We recorded the number of cycles until failure and the type of failure. Implant failure was analyzed by visual inspection and scanning electron microscopy. Open circuit potential and potenctiodynamic tests were carried out with high precision potentiostat using Hank’s solution at 37 °C to evaluate the effect of the implantoplasty on the corrosion resistance. Implantoplasty significantly reduced the surface topography values (median) and interquartile range (IQR); S_a_ from 1.76 (IQR = 0.11) to 0.49 (IQR = 0.16), S_z_ from 20.98 (IQR = 8.14) to 8.19 (IQR = 4.16), S_sk_ from 0.01 (IQR = 0.34) to −0.74 (IQR = 0.53) and S_dr_ from 18.20 (IQR = 2.26) to 2.67 (IQR = 0.87). The fatigue limits of the control and implantoplasty groups were 551 N and 529 N, respectively. The scanning electron micrographs showed fatigue striations indicating fatigue failure. Besides, the fractographic analysis revealed a typical brittle intergranular fracture mechanism. The infinite life range of the dental implants evaluated was largely above the threshold of usual chewing forces. Implantoplasty seems to render a fairly smooth surface and has a limited impact on fatigue resistance. In addition, implantoplasty produces a decrease in the corrosion resistance of the implant. Corrosion current density from 0.019 μA/cm^2^ for as-received to 0.069 μA/cm^2^ in the interface smooth-roughened dental implant. These places between the machining and the rough area of the implant are the most susceptible, with the appearance of pitting.

## 1. Introduction

Dental implants are a predictable long-term treatment option for the esthetic and functional rehabilitation of patients with partial or total edentulism [1,2]. However, different complications can arise and jeopardize the results of implant-prosthetic rehabilitation [3]. Peri-implant diseases (both peri-implant mucositis and peri-implantitis) are considered to be the most common long-term complications associated with dental implants. Clinically, inflammatory peri-implant diseases are categorized into peri-implant mucositis or peri-implantitis. In the 1st European Workshop on Periodontology (EWOP), peri-implant mucositis was defined as a reversible inflammatory reaction in the soft tissues surrounding a functioning implant, and peri-implantitis was described as inflammatory reactions associated with loss of supporting bone around a functioning implant [2]. Both disorders are associated with an inflammatory reaction caused by bacterial biofilm, affecting an osseointegrated dental implant [4]. Peri-implantitis is characterized by inflammatory changes in the peri-implant mucosa and progressive bone loss [5].

Nonsurgical treatment has been shown to offer limited efficacy in the remission of peri-implantitis. Antibiotic treatments are not very effective in the long term since drug release can last up to a maximum of two weeks. In addition, when the biofilm is formed, it is difficult to attack the bacteria because there is a protective effect that is very difficult to eliminate. That is why it is necessary to resort either to a change of dental implant or to try implantoplasty. [6,7,8,9]. A possible reason could be insufficient decontamination of the implant surface, which is exposed to bacterial colonization and is usually moderately rough. In fact, the macro-geometry of the threads and the surface roughness of the implant can further complicate decontamination in the presence of associated peri-implant bone loss [10]. Depending on the morphology and the extent of the defect, as well as on the location of the implant, surgical treatment can involve different approaches and implant surface decontamination techniques [11]. Among these techniques, implantoplasty (IP) consists of polishing and smoothing those parts of rough-surfaced implants that are outside the bone contour due to progressive marginal bone loss associated with peri-implantitis, or eventual bone resection during peri-implant surgery [12]. Although this technique has proved effective in clinical studies [13,14], several investigations have reported that IP reduces the fracture resistance of both standards (i.e., 3.75 to 4.5 mm) and narrow (i.e., ≤3.5 mm) diameter implants [15,16,17,18]. However, none of these studies have determined the fatigue limit in an unfavorable clinical scenario (i.e., narrow implants with a diameter ≤3.5 mm and bone loss equivalent to 50% of their length). These situations are common in dental implants that require IP.

Thus, the primary objective of the present study was to determine the effect of IP upon the fatigue resistance of Ti6Al4V narrow screw-shaped dental implants, with internal connection and a moderately rough surface, in the presence of 50% bone loss and the corrosion resistance of the implants treated.

## 2. Materials and Methods

### 2.1. Dental Implants

Twenty tapered, screw-shaped Ti6Al4V (titanium grade 5) commercial dental implants were tested in the present in vitro study (Biomimetic Ocean^®^ 3.5 mm wide and 10 mm long with internal hexagonal connection, Avinent^®^ Implant System, Santpedor, Spain) (Figure 1). The surface was moderately rough after sandblasting by abrasive particles (Al_2_O_3_), acid-etching, and anodization process. The roughness of the implant presents a Ra of 0.9 μm. We used a computer-generated random sequence to allocate 10 implants per group, and subsequently performed implantoplasty of the implants in the IP group.

### 2.2. Implantoplasty Procedure

Implantoplasty was performed following the simplified three-bur protocol described by Costa-Berenguer et al. [19]. We inserted a cover screw to protect the implant connection from titanium debris and removed the threads of the coronal half of the implants using an oval-shaped tungsten carbide bur (H379.314.023 Komet, GmbH and Co. KG, Lemgo, Germany) with an air-driven high-speed handpiece under water irrigation. We instructed the operator to limit the initial steps of implantoplasty to the threads of the dental implant, as recommended by Schwarz et al. [12]. Then, we polished the resulting surface with two silicon carbide polishers and the same handpiece (9618.314.030 and 9608.314.030 Komet, GmbH and Co. KG, Lemgo, Germany) (Figure 2).

Implantoplasty was performed on two consecutive days by an experienced clinician (O.C-F.) that had been involved in previous studies with a similar design. Magnification loupes (2.8×) with a LED light (Galilean HD and Focus™ LED 6000 k, ExamVision ApS, Samsø, Denmark) were used and implantoplasty was conducted until the 5-mm coronal portion of the implant exhibited a uniformly smooth and shiny surface. The pressure applied and the number of strokes was not standardized in order to increase the external validity of the study. A new set of burs or tips was used for every other implant. After IP, the implants were thoroughly cleaned by irrigation using distilled water and dried with compressed air. Finally, the cover screw was removed.

### 2.3. Surface Topography Analyses

All samples were analyzed with a confocal optical microscope (Leica^®^ DCM 3D, Leica Microsystems AG, Wetzlar, Germany) under 20× magnification and a numerical aperture of 0.50. Three regions of interest of 636 × 442 µm were determined: immediately below the smooth surface of the platform (T0), at 2.5 mm (T2.5), and at 5 mm (T5) from the platform in the apical direction. The LeicaMap^®^ software (Leica Microsystems AG, Wetzlar, Germany) was used to measure the surface topography and to calculate the surface roughness parameters.

The surface roughness of each area was defined using the following normalized three-dimensional parameters:

S_a_ (arithmetic mean height) is defined as the difference in height of each point compared to the arithmetical mean of the surface.

S_z_ (average maximum height) is defined as the sum of the largest peak height value and the largest pit depth value within the defined area.

S_sk_ (skewness of topography height distribution) is defined as the degree of bias of the roughness shape.

S_dr_ (developed interfacial area ratio) is defined as the ratio between the area of the “real” developed surface and the area of the “projected” surface.

Form was previously removed, and a Gaussian filter of 30 μm was applied for roughness and waviness. Only roughness parameters were assessed. The values of the roughness are shown in Table 1.

### 2.4. Cast Preparation

In a second step, we embedded the implants in the same position using resin casts, in such a way that 5 mm of the rough surface was exposed. This approach was chosen to simulate horizontal bone resorption of 5 mm (50% of the total implant length), which is 2 mm more than the International Standardization Organization (ISO) 14801:2016 specifications. The epoxy resin was EA 3471 A and B Loctite^®^ (Henkel AG and Company, Düsseldorf, Germany) to simulate bone (Young’s modulus of elasticity ≥ 3 GPa). In Figure 3 can be observed the preparation of the samples.

### 2.5. Fatigue Testing

We carried out fatigue testing in room air and at room temperature using a servo-hydraulic mechanical testing machine (MTS Bionix 370, MTS^®^, Eden Prairie, MN, USA) equipped with a 15 kN load cell (MTS Load Cell 661.19H-03, MTS^®^, Eden Prairie, MN, USA). We screwed identical hemispherical abutments to each implant with the torque recommended by the manufacturer (35 N·cm). The loading center was located 13 mm above the resin (nominal bone level). According to ISO 14801:2016, we placed the samples in a stainless-steel clamping jaw so that loading had an angle of 30° to the longitudinal axis of the implant (Figure 4).

According to European Standard EN ISO 14801:2016 (Dentistry–Implants–Dynamic loading test for endosseous dental implants), the general principles for fatigue testing state that “at least two, and preferably three, specimens shall be tested at each of at least four loads. Moreover, “at least three specimens shall be tested, and every specimen shall reach the specified number of cycles with no failures” in order to reach the infinite life range. For all these reasons, a minimum of 9 specimens are necessary (in our study there were 10 samples in the experimental group and 9 in the control group) to meet the requirements of the International Standard.

To conclude, it should be noted that while this International Standard simulates de functional loading of an endosseous dental implant under “worst case” conditions, it is not applicable for predicting the in vivo performance of an endosseous dental implant or dental prosthesis, particularly if multiple endosseous dental implants are used for a dental prosthesis.

Each specimen received a maximum of 5,000,000 cycles of a uniaxial load, perpendicular to the tangent of the dome of the hemispherical abutment. Loading range was between a maximal nominal value and 10% of this value (R = 0.1). To minimize the vibrations of the testing machine, the sinusoidal load frequency was kept at 15 Hz. We used TestStar II^®^ software (MTS^®^, Eden Prairie, MN, USA) to record data in real-time.

In accordance with ISO 14801:2016, tests were carried out applying a minimum of four series of loads, the first of which was equivalent to 80% of the maximum compression force (F_maxC_ and F_maxIP_), which was determined in a previous study to be 735 N and 529 N for the control and IP samples, respectively 21. At each load level, two samples were evaluated, considering 5 × 10^6^ cycles as an infinite life criterion. If any of the samples collapsed before reaching the specified number of cycles, the procedure was started again with two new implants and under a lesser load (20% if ≥60% Fmax and 10% if <60% Fmax). When two consecutive samples reached 5 × 10^6^ cycles without failure, an additional test was performed with a third sample. If the latter did not fail (i.e., 3 consecutive samples without apparent failure), this point was considered to be the fatigue limit beyond which the implant could withstand an infinite number of loading cycles. In case the fatigue limit was reached in less than four load series, additional levels (1, 2, or 3) were established by applying a load 5% higher than the previous one. The number of cycles and the state (i.e., intact or failed) of each tested specimen was recorded. Failure was defined as the elastic limit of the material, permanent deformation, loosening of the implant assembly, or fracture of any component.

Additionally, for the maximal supported load, the maximal bending moment (M) was calculated using Equation (1):(1)M=F×l×sin30°
where *l* is the distance (in cm) from the center of the load hemisphere to the nominal bone level and *F* (in N) is the maximal supported load.

The results of the fatigue tests were displayed in a load versus the number of cycles plot (i.e., S-N curve or Wöhler’s curve), which represents the number of load cycles of each sample (logarithmic scale) and the corresponding maximal load (linear scale).

### 2.6. Fractographic Analysis

All failed specimens were assessed by visual inspection and SEM (Quanta−200, Field Electron and Ion Company, Hillsboro, OR, USA) to describe the failure pattern.

### 2.7. Corrosion Tests

A total of 10 samples, 5 as-received and 5 treated, were used for the corrosion tests. The test area for each sample was 15.2 mm^2^. The electrolyte for all tests was Hank’s solution (ThermoFisher, Madrid, Spain) (Table 2). Corrosion will be localized on the treated implants at the interface of the roughness and the machined part since it is in that area that differences in residual stresses and topography will produce corrosion potentials. It is for this reason that we are going to study this area of treated dental implants [20,21].

The electrochemical cell used can be observed in Figure 5. For both the open circuit potential and the potentiodynamic tests, the reference electrode used was a calomel electrode (saturated KCl), with a potential of 0.241 V. All tests were performed at 37 °C inside a Faraday box to inhibit the external electric or electromagnetic fields.

Open-circuit potential tests were carried out for 5 h for all the samples, taking measurements every 10 s. The potential was stabilized when the variation of the potential is lower than 2 mV for 30 min according to the ASTM G31 standard [22,23,24]. The data and the E-t curves were obtained using the PowerSuite software with the PowerCorr-Open circuit. Cyclic potentiodynamic polarization curves were determined according to the ASTM G5 standard [23]. The counter electrode used was platinum [24]. After stabilization, the potentiodynamic test was launched, performing a cyclic sweep from −0.8 mV to 1.7 mV at a speed of 2 mV/s. These parameters were entered into the PowerSuite program using the PowerCorr-Cyclic Polarization function to obtain the curves. The parameters studied were: i_corr_ (μA/cm^2^)/corrosion current density. E_corr_ (mV)/Corrosion potential: value at which the current density changes from cathodic to anodic.

The E_corr_ and i_corr_ parameters are obtained by extrapolating the Tafel slopes. In accordance with the ASTM G102–89 standard [24], these values are then used to calculate the polarization resistance (R_p_) using the Stern-Geary expression and the corrosion rate (CR in mm/year) [24,25,26,27,28].

### 2.8. Statistical Analysis

Categorical variables were reported as absolute and relative frequencies. We explored the normal distribution of scale variables (roughness parameters) with the Shapiro-Wilk’s test and visual analysis of the P-P and box plots. The mean and standard deviation (SD) were calculated and, if a normal data distribution was ruled out, the median and interquartile range (IQR) were calculated. The Mann-Whitney U-test was used to compare the groups. The statistical analysis was carried out with the Stata14 statistical package (StataCorp^®^, College Station, TX, USA) at a level of significance *p* < 0.05.

## 3. Results

### 3.1. Surface Topography Analyses

The surface topography results are shown in Table 2. The median S_a_ (arithmetic mean height), S_z_ (average maximum height), S_sk_ (skewness of topography height distribution), and S_dr_ (developed interfacial area ratio) values of the IP group were significantly lower than those of the control group (*p* ≤ 0.001) (Table 2). In the treated surfaces, the roughness of the coronal, middle, and apical areas was similar and no statistically significant differences were found in any of the parameters (S_a_, S_z_, S_sk,_ and S_dr_). Representative confocal microscope 3D topography images of the control and experimental samples for each of the three regions of interest (T0, T2.5 and T5) are depicted in Figure 6.

### 3.2. Fatigue Testing

Nineteen implants underwent fatigue testing: 10 implants in the IP group and 9 implants in the control group (Table 3). Three consecutive samples subjected to IP withstood the 5 × 10^6^ cycles of the initial load level without apparent damage. This was equivalent to 529 N, which corresponds to 80% of the maximum compression force (F_maxIP_). The fatigue limit of the control group was 551 N (i.e., 60% of F_maxC_) (Table 3).

Considering that the distance between the nominal bone level and the center of the hemispherical load abutment was 1.3 cm, the maximum bending moments (M) were:(2)$$M_{IP}=1.3\;cm\;\sin30\;529N=343.85\;Ncm$$
(3)$$M_{C}=1.3\;cm\;\sin30\;551N=358.15\;Ncm$$

Load versus the number of cycles (S-N curves) in the IP and control groups is represented in Figure 7. Two different regions could be identified: (1) the finite life region was found above 551 N (i.e., 60% of F_maxC_); and (2) the infinite life range which started below that threshold. Similarly, in the load versus the number of cycles plot obtained for the IP samples (Figure 7), we determined: (1) a transition region above 529 N (i.e., 80% of F_maxIP_); and (2) an infinite life range that started below that threshold.

All failed samples exhibited a fracture pattern perpendicular to the longitudinal axis of the implant in a region of the implant body close to the embedding plane (Table 3). This area is the least thick zone, due to the presence of the hollow space for the prosthetic screw.

The micrographic analysis of the fracture surface revealed a typical brittle intergranular fracture mechanism with secondary cracking. In all cases, fatigue failures started at the implant body. More specifically, the fracture began on the side of the implant subjected to continuous and oscillating stresses. Accumulated damage led to rupture on exceeding the mechanical resistance of the material (Figure 8). Fatigue cracks are always nucleated on the surface of the dental implant, as can be seen in Figure 8. The location of the breakage is at the screw connection as can be seen since this is the area with the smallest cross-section of the dental implant. The area that is observed smoother is the area of crack propagation and subsequently the ductile fracture of the material since the resistant section is reduced as the crack progresses.

For the corrosion studies, the results can be observed in Table 4. These results show that the highest open-circuit corrosion potential values (E_OCP_) were obtained for as-received dental implant.

The potentiodynamic analysis confirmed that the treatment that produced surfaces with the best corrosion resistance was as-received dental implants showing the lowest values of corrosion current density (i_corr_) and corrosion rate (V_c_). In addition, the original implants show the highest resistance to polarization (R_p_). Implantoplasty produce a loss of the corrosion resistance with respect to the as-received samples. Figure 9 shows pitting in the interface area of the rough part of the dental implant and in the smoother area. This is the zone of the greatest potential difference generated by the differences in internal stress of the material between the rough and the machined part. It is well known that the energetic heterogeneities on the surface are points susceptible to corrosion [27,28].

## 4. Discussion

The present in vitro study assessed the reduction of fatigue strength of narrow-diameter dental implants with internal hexagonal connection in a model that simulated a horizontal peri-implant defect equivalent to 50% of the implant length. To the best of our knowledge, this is the first study to analyze the effect of IP in this worst-case scenario. Implantoplasty had a limited influence on the fatigue limit and these values were always above the usual chewing forces [29,30]. It is important to stress that implantoplasty allowed to significantly reduce the roughness of the implants and to create a minimally rough, groove-free surface [31]. Regardless of the protocol used to polish the dental implant, implantoplasty allows a reduction of the implant surface roughness without compromising its biocompatibility [12,32,33,34,35,36,37,38]. However, our results have shown that it was not possible to achieve a completely smooth surface (Sa > 0.30 µm) in all cases, which is in line with the finding published by Beheshti Maal et al. [34].

In our study, IP was carried out under conditions that simulated the real-life clinical scenario [13,39,40], though less challenging. As previously reported with a similar protocol 31, smooth areas might be difficult to achieve in clinical practice, particularly in locations with difficult access or when it is not possible to remove the prosthesis. This might result in more aggressive thinning of the implant walls and, consequently, poorer mechanical properties than those reported herein.

One of the major concerns related to IP is the mechanical behavior of the dental implant after polishing [33]. In this study, the researcher was instructed that IP should be limited to the threads of the implant as suggested by Schwarz et al. [12] but the procedure was not fully standardized with the intention of increasing the external validity of the results. This might be considered a limitation since performing a manual IP implies a source of variation that could lead to a heterogeneous reduction of the implant walls. Indeed, a reduction of the thickness of the implant walls is to be assumed [8,41,42,43,44,45,46]. More precisely, based on the results of an investigation using the same protocol, a homogeneous and constant reduction in width of 0.13 mm (CI 95%: 0.06 to 0.19) throughout the fixture should be expected [41].

Aside from the macroscopical changes, other factors can also influence the mechanical behavior of dental implants: the implant material [47], the implant-abutment connection design [48,49], implant diameter [15,18,50,51], crown to implant ratio [43], crown height [52], and nominal bone level 34. This is the reason why we selected a worst-case scenario involving commercially pure titanium narrow diameter (<3.5 mm) implants, with an internal connection and thin walls, an unfavorable crown-to-implant ratio, and with a significant loss of supporting bone.

In the present study, six control implants fractured, whereas only three implants with implantoplasty did not withstand the 5 × 10^6^ loading cycles. This finding could be explained by the fact that the control samples were submitted to higher initial loading levels in comparison with the experimental implants (735 N vs. 529 N), according to the results of the previously performed static load tests 31. Hence, monotonic loading seems to have a reduced clinical relevance as mechanical failures are more likely after the application of repeated loads [53]. It should be also stressed that, according to Shemtov-Yona et al. [54], narrow implants have shown an unpredictable fatigue behavior. In this sense, only two different regions could be identified in the S-N curves (Figure 7)

In a previous study [41], we performed compression tests and found IP to significantly reduce fracture resistance (*p* < 0.001). Specifically, the external hexagonal, internal hexagonal and internal conical connection groups exhibited a decrease in Fmax of 27.96%, 28.00%, and 29.41%, respectively [41]. However, the clinical relevance of these static loading tests is limited because factors such as time or environment are not taken into account [55]. In fact, mechanical failure of dental materials usually occurs once they withstand repeated cycles of low-energy stress, rather than higher static loads [56]. Three in vitro studies have analyzed the effect of cyclic loading upon the fracture resistance of implant materials [18,45,57]. However, to the best of our knowledge, the present study is the first to determine the effect of IP upon the dental implant fatigue limit. The application of >1 × 10^6^ cyclic loads reduces fracture strength by introducing a “mechanical aging” effect in the tested components [45,57]. Our results suggest that even in this worst-case scenario (i.e., 3.5 mm diameter internal hexagonal connection implant, with bone loss equivalent to 50% of its length and subjected to IP), implants showed an infinite life range above 500 N, which is well above the threshold of the forces recorded during chewing and swallowing (around 250 N) [19,20].

The fractographic analysis revealed a classical fatigue failure pattern, with the presence of fatigue striations, perpendicular to the fatigue crack direction, in all fractured specimens. The scanning electron microscope (SEM) images are consistent with the results obtained in clinical studies that analyze implant fractures [58]. As expected, the most susceptible area to fracture was the implant body near the embedding plane probably due to the presence of the hollow space for the prosthetic screw. As published previously, external hexagonal connection implants might be more resistant, and this might change the fatigue fracture pattern 31. Future studies should assess if the present results can be extrapolated to other implant-abutment connection designs.

Several IP protocols have been described in the literature. Most publications agree that IP, regardless of the burs used, significantly reduces roughness [32]. This could have an impact upon peri-implant health, as the composition and development of biofilms on the surface of dental materials correlate with their surface roughness and free surface energy [59]. Other physicochemical properties, such as surface charge or substrate stiffness, appear to be of lesser importance [60,61]. Regarding the surface roughness of the dental implant, Ra and Sa are the most appropriate parameters for predicting susceptibility to bacterial adhesion [62]. In fact, roughness has no influence upon bacterial adhesion at R_a_ < 0.2 µm [63,64]. Although our IP protocol resulted in a minimally rough surface (S_a_ = 0.49 µm), this still might not be smooth enough to impede bacterial adhesion. Thus, even if complete decontamination of the surface is achieved with IP, bacterial recolonization will occur within a short period of time [23,26,27]. Therefore, it is crucial for the prosthesis to facilitate hygiene of the treated area and, at the same time, for the patient to maintain good plaque control. Furthermore, the patient should follow a maintenance program with follow-up visits at least every 4–6 months in order to avoid reinfection or the recurrence of peri-implantitis [64].

Implantoplasty removes biofilms from the titanium surface to prevent peri-implantitis and preserves the osseointegrated dental implant. However, as we have seen, it reduces the resistance to corrosion. As mentioned above, the interface zone is where the greatest potential for corrosion is generated and is the most susceptible to corrosion as we have seen in Figure 9. In general, when these differences between surface stresses occur in a metal, an annealing heat treatment is performed to release the internal stresses. Obviously, this heat treatment cannot be performed in the patient’s mouth and therefore it is necessary to investigate a new passivation treatment to be performed in vivo. This passivation must not cause damage to the surrounding tissues and create a titanium oxide layer that generates the passivation of the dental implant to avoid electrochemical corrosion [65]. On the other hand is very important the mouthwashes and hygiene. Bianchi et al. [66] studied different mouthwashes with fluoride compounds and they confirm that the longer permanence of the products may lead to a more effective plaque control. It would also be desirable that this passivation treatment could promote cell adhesion to biologically regenerate the implant and achieve a new re-osseointegration.

It should be noted that the present study has a number of limitations, one being the design of the dental implant as well as the material from which the dental implant was fabricated, in our study Ti6Al4V. However, there are many dental implants fabricated from commercially pure Ti which has lower mechanical properties than the alloy, and therefore the influence of implantoplasty on fatigue and corrosion may be different. Another limitation of the study is the clinician who performs the implantoplasty, although a protocol is rigorously followed, there are aspects that cannot be controlled, such as the tension applied on the metal with the drills, and orientation... all these variations can cause alterations in the results of this research work.

## 5. Conclusions

The infinite life range of the evaluated dental implants was largely above the threshold of usual chewing forces, with the fatigue limit of the implantoplasty group being 529 N. Thus, implantoplasty does not seem to significantly reduce fatigue resistance even in unfavorable situations where narrow-diameter internal hexagonal connection implants are involved. With carbide burs and silicon carbide polishers, Sa values < 0.5 μm can be obtained. There is a significant decrease in the corrosion resistance of the treated dental implants, especially at the smooth-rough interface of the dental implant, where pitting was observed. Further studies are required to determine whether these results are achievable in a real-life clinical setting.

## Figures and Tables

**Figure 1 jfb-14-00061-f001:**
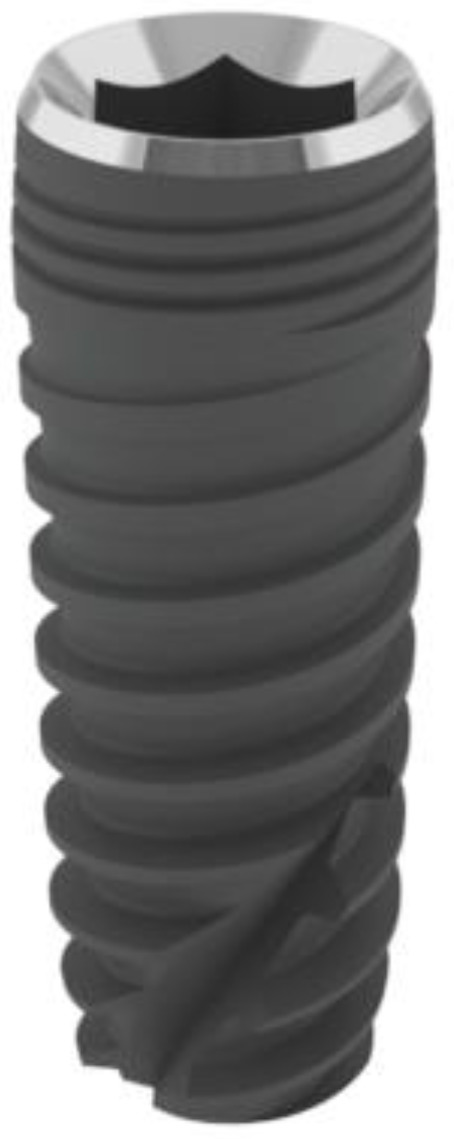
Dental implant used in this study.

**Figure 2 jfb-14-00061-f002:**
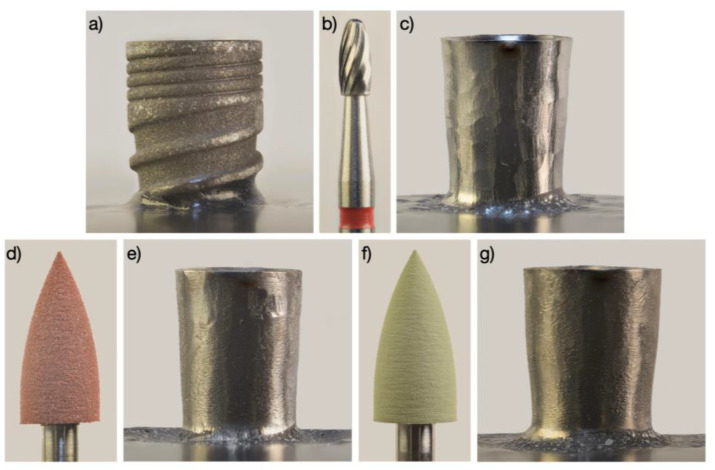
Simplified three-bur protocol IP procedure. (**a**) Macroscopic appearance of the implant; (**b**) Tungsten carbide bur; (**c**) Macroscopic appearance of the implant after applying the tungsten carbide bur; (**d**) Brown silicon carbide polisher; (**e**) Macroscopic appearance of the implant after applying the brown silicon carbide polisher; (**f**) Green silicon carbide polisher; (**g**) Macroscopic appearance of the implant after applying the green silicon carbide polisher.

**Figure 3 jfb-14-00061-f003:**
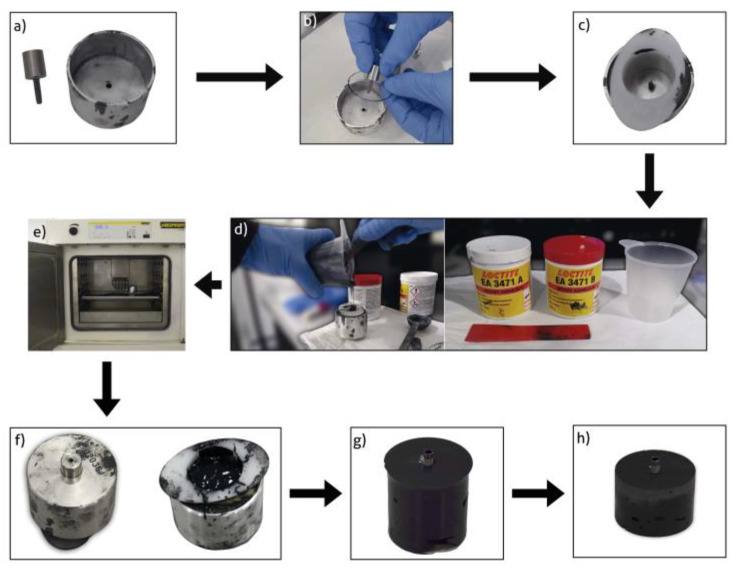
Process of making the study specimens. (**a**) Holding the implant in the cartridge. Metal tray; (**b**) Insertion of the implant in the tray and placement of the separator; (**c**) Placement of the plastic mould; (**d**) Mixing of epoxy resins; (**e**) Baked; (**f**) Specimen once baked; (**g**) Unmoulded; (**h**) Standardization of dimensions.

**Figure 4 jfb-14-00061-f004:**
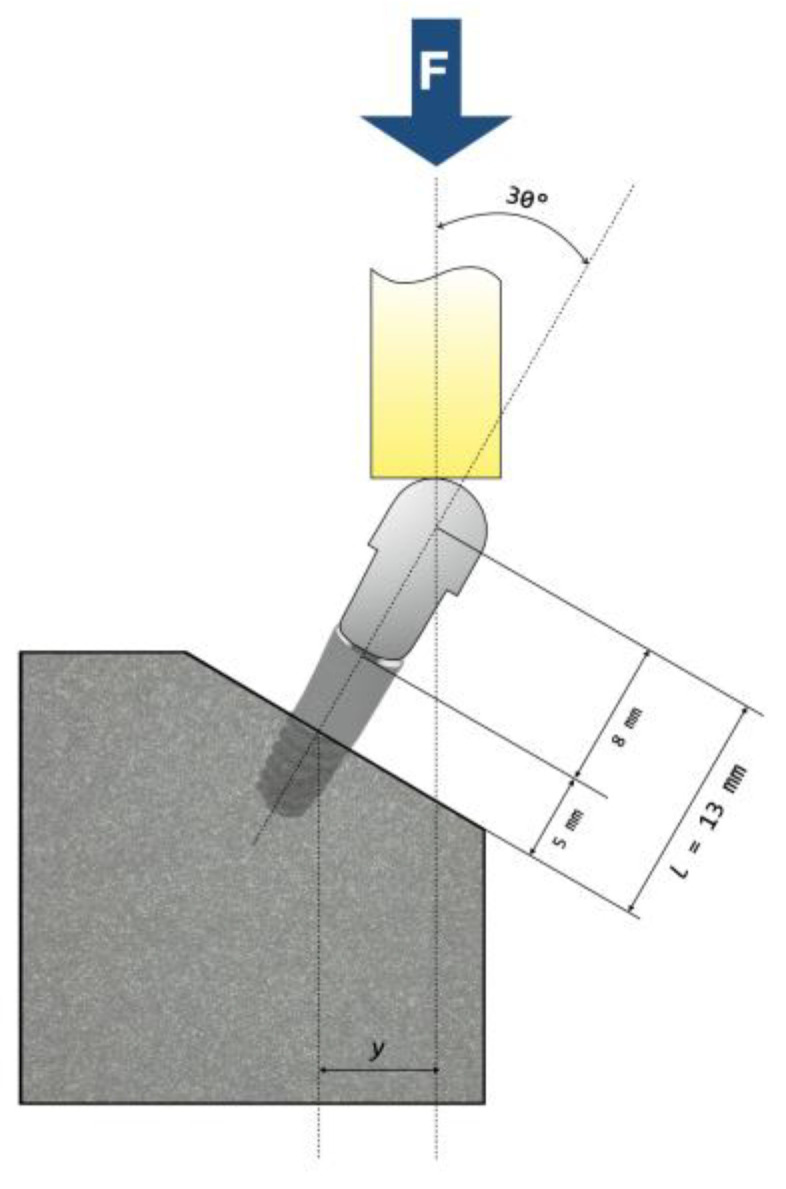
Schematic representation of the test setup according to ISO 14801:2016, except for bone nominal level.

**Figure 5 jfb-14-00061-f005:**
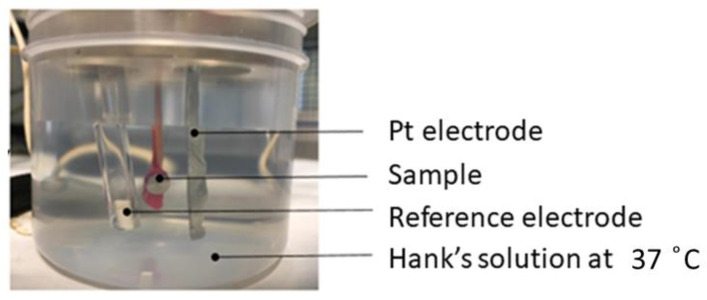
Experimental setup used for assessing corrosion resistance.

**Figure 6 jfb-14-00061-f006:**
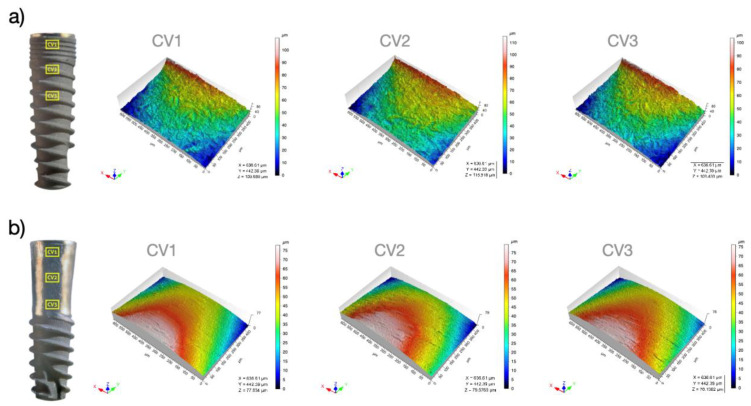
Confocal microscope 3D surface topography analyses of the control (**a**) and experimental samples (**b**) for each of the three regions of interest (CV1–3).

**Figure 7 jfb-14-00061-f007:**
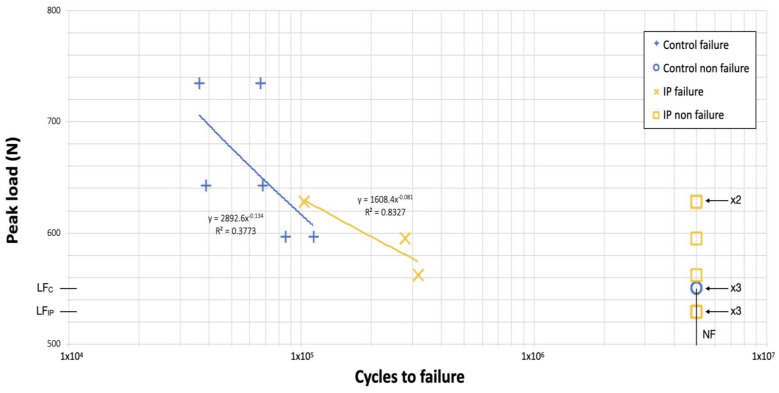
Load versus the number of cycles plot (S-N curve) for each group.

**Figure 8 jfb-14-00061-f008:**
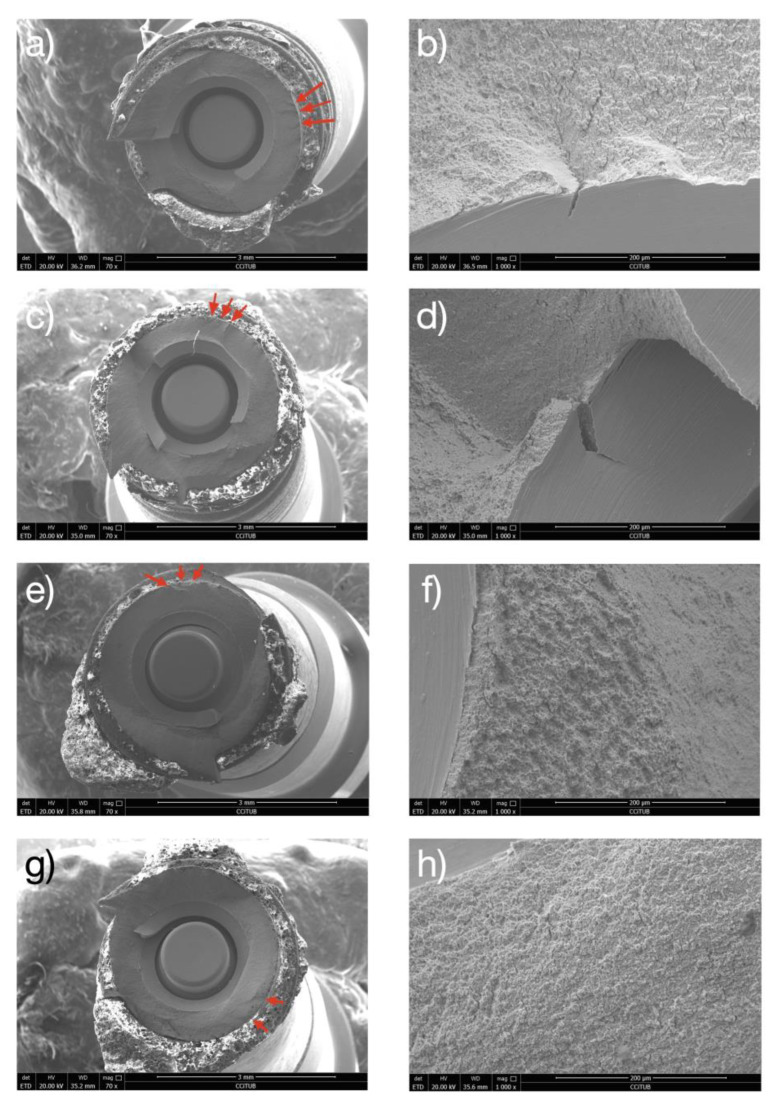
Fractographic analysis with scanning electron microscopy. (**a**–**d**) Control samples, (**a**,**c**) are macroscopic aspect of the fractured dental implants where the red arrows indicate the crack nucleation in the implant surface. (**b**,**d**) are the images at more magnification of (**a**,**c**) where can be observed the crack and the secondary cracks and their propagation. (**e**–**h**) Test samples. (**e**,**g**) are macroscopic images where the red arrows indicate the crack nucleation. (**f**,**h**) are image of the fracture surface where can be observed the ductility of the titanium.

**Figure 9 jfb-14-00061-f009:**
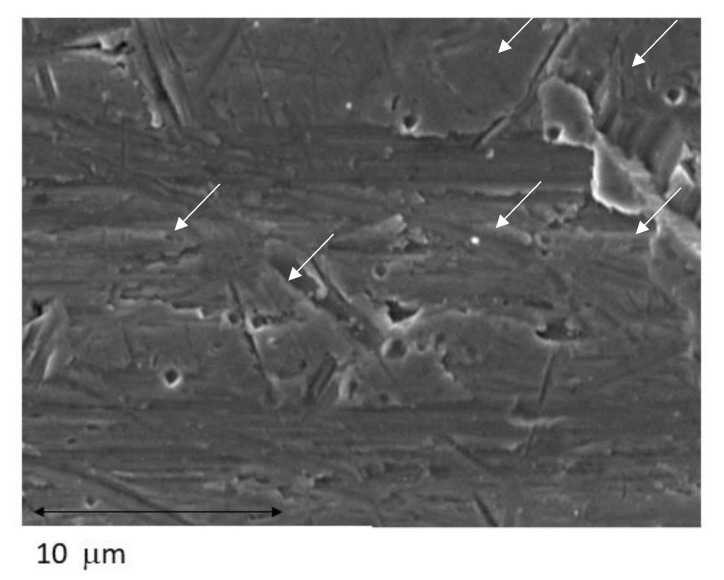
Pitting observed by SEM in the interface smooth-roughness of the dental implant treated. (Arrows show the pitting in the titanium).

**Table 1 jfb-14-00061-t001:** Median surface roughness.

	S_a_ (µm)		S_z_ (µm)		S_sk_		S_dr_ (%)	
Sample	Control	IP	Control	IP	Control	IP	Control	IP
1	1.88	0.47	20.86	13.34	0.29	−1.02	18.95	2.62
2	1.84	0.49	19.60	6.20	−0.04	−0.46	18.97	2.57
3	1.82	0.58	30.64	7.66	0.31	−0.86	18.73	2.61
4	1.78	0.44	23.84	7.32	−0.03	−0.78	18.93	2.42
5	1.73	0.65	21.11	15.76	−0.07	−0.70	17.67	5.01
6	1.87	0.50	33.41	10.47	0.64	−0.99	20.96	3.45
7	1.72	0.33	20.15	5.04	−0.05	−1.39	16.29	1.19
8	1.71	0.73	19.21	8.72	0.05	−0.14	16.69	3.84
9	1.74	0.61	19.78	9.09	−0.14	−0.71	16.35	2.95
10	1.72	0.34	27.92	6.31	0.05	0.18	16.89	2.75
Median	1.76	0.49	20.98	8.19	0.01	−0.74	18.20	2.67
IQR	0.11	0.16	8.14	4.16	0.34	0.53	2.26	0.87
*p-value*	<0.001 *		<0.001 *		0.001 *		<0.001 *	

Abbreviations; IQR: Interquartile range; IP: Implantoplasty. * Significant association (*p* < 0.05).

**Table 2 jfb-14-00061-t002:** Composition of Hank’s solution.

Chemical Product	Composition (mM)
K_2_HPO_4_	0.44
KCl	5.4
CaCl_2_	1.3
Na_2_HPO_4_	0.25
NaCl	137
NaHCO_3_	4.2
MgSO_4_	1.0
C_6_H_12_O_6_	5.5

**Table 3 jfb-14-00061-t003:** Results of cycling tests.

% F_max_ Total	Peak Load (N)	Number of Cycles	Failure	
			Location	Description
Implantoplasty group (*n* = 10)				
95%	628	5,000,000	Absence of failure	
95%	628	5,000,000	Absence of failure	
95%	628	102,360	Implant body	Fracture
90%	595	279,251	Implant body	Fracture
90%	595	5,000,000	Absence of failure	
85%	562	318,799	Implant body	Fracture
85%	562	5,000,000	Absence of failure	
80%	529	5,000,000	Absence of failure	
80%	529	5,000,000	Absence of failure	
80%	529	5,000,000	Absence of failure	
Control group (*n* = 9)				
80%	735	36,364	Implant body	Fracture
80%	735	66,690	Implant body	Fracture
70%	643	38,830	Implant body	Fracture
70%	643	68,519	Implant body	Fracture
65%	597	112,841	Implant body	Fracture
65%	597	85,644	Implant body	Fracture
60%	551	5,000,000	Absence of failure	
60%	551	5,000,000	Absence of failure	
60%	551	5,000,000	Absence of failure	

Abbreviations; F_max_: Maximum compression force.

**Table 4 jfb-14-00061-t004:** Electrochemical and corrosion parameters assessed for dental implants.

Samples	E_OCP_ (mV)	i_corr_ (μA/cm^2^)	E_CORR_ (mV)
As-received	−141.7 ± 0.3	0.019 ± 0.010	−380 ± 18
Interface smooth-roughned	−194.3 ± 0.4	0.069 ± 0.016	−450 ± 40

## Data Availability

The data presented in this study are available on request from the corresponding author. All authors are willing to share our research data.

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
