# Peer review of "Effect of Implantoplasty on Roughness, Fatigue and Corrosion Behavior of Narrow Diameter Dental Implants"

_jfb, 2023, doi:10.3390/jfb14020061_

Round 1
Reviewer 1 Report
This exquisitely designed research have demonstrated that implantoplasty does not significantly reduce the fatigue resistance of narrow-diameter internal hexagonal connection implants. The conclusions of the study are instructive for clinical process. The paper is well written, it clearly expresses the corrosion resistance and fatigue resistance of implants.
There are some problems, which must be solved before it is considered for publication.
Fisrtly, in page 1, line166 and 169, the format of digits is incorrect. Please check the manuscript carefully for other incorrectly typed digits.
Secondly, the authors mentioned in page 12, line 347 that a reduction of the thickness of the implant walls is to be assumed and different executer of manual implantoplasty may possibly lead to heterogeneous reduction. As far as I am concerned, instruments being used in implantoplasty varies from a practitioner to another. It is probably safer to draw conclusions with the limitations of the mean reduced volume of the thickness of implant wall.
Thirdly,There is at least one spelling error in the manuscript, such as, in page 2,line 69, “bone lossanf” would be “bone loss and”. Please check the manuscript carefully.
Author Response
REVIEWER 1
Dear Reviewer,
Thanks for taking the time to review our manuscript and suggest to us to improve our work by providing a lot more detail. We have done so, and we are now submitting a manuscript that not only addresses the points you specifically raised but also many others that we have considered in order to deliver what we think is a much-improved version of our work. This version includes more paragraphs, figure, English grammar revisions in all main sections, new references. Thanks a lot. We are looking forward to your comments.
Sincerely,
Francisco-Javier Gil Mur
This exquisitely designed research have demonstrated that implantoplasty does not significantly reduce the fatigue resistance of narrow-diameter internal hexagonal connection implants. The conclusions of the study are instructive for clinical process. The paper is well written, it clearly expresses the corrosion resistance and fatigue resistance of implants.
There are some problems, which must be solved before it is considered for publication.
- Fisrtly, in page 1, line166 and 169, the format of digits is incorrect. Please check the manuscript carefully for other incorrectly typed digits.
Done. Five digits have been corrected.
- Secondly, the authors mentioned in page 12, line 347 that a reduction of the thickness of the implant walls is to be assumed and different executer of manual implantoplasty may possibly lead to heterogeneous reduction. As far as I am concerned, instruments being used in implantoplasty varies from a practitioner to another. It is probably safer to draw conclusions with the limitations of the mean reduced volume of the thickness of implant wall.
Thank you for your important comment. The authors have introduced in the discussion a new paragraph with the limitations of this stuy according to the reviewer.
- Thirdly,There is at least one spelling error in the manuscript, such as, in page 2, line 69, “bone lossanf” would be “bone loss and”. Please check the manuscript carefully.
Revised
Reviewer 2 Report
1. Introduction: there is no details about experimental hypothesis nor the clinical significance behind this study. Why the authors aimed to study roughness, fatigue, and corrosion resistance. This is not clear to the readers
2. Methodology:
- Was there a power test to estimate the required number of samples in both groups. It is not shown.
- No information or details provided about type of the implant, characteristics, surface
- The distribution of samples in both groups is not clear especially procedures related to the control group
- Was the implant surfaces roughened by the operators or were originally rough and how the level of roughness decided
- The IP procedure was carried out before mounting the implants in the resin cast. This is far from what is done clinically or at least after mounting. Although some steps were taken to avoid aggressive threats removal but the direct access and close manual manipulation does not resemble clinical settings. This is a major flaw in the design
-No appropriate citations were made for any IP procedure nor for roughness, fatigue and corrosion
- Figure 2 legends are not adequate with some text missing. Burs mixed up with implants
3. Results
- It is not clear in the figure of fractography what regions or levels appear
.4. Study limitations should be added at the end
5. Language editing is needed, many typos
Author Response
REVIEWER 2
Dear Reviewer,
Thanks for taking the time to review our manuscript and suggest to us to improve our work by providing a lot more detail. We have done so, and we are now submitting a manuscript that not only addresses the points you specifically raised but also many others that we have considered in order to deliver what we think is a much-improved version of our work. This version includes more paragraphs, figure, English grammar revisions in all main sections, new references. Thanks a lot. We are looking forward to your comments.
Sincerely,
Francisco-Javier Gil Mur
- Introduction: there is no details about experimental hypothesis nor the clinical significance behind this study. Why the authors aimed to study roughness, fatigue, and corrosion resistance. This is not clear to the readers
- Methodology:
- Was there a power test to estimate the required number of samples in both groups. It is not shown.
According to European Standard EN ISO 14801:2016 (Dentistry – Implants – Dynamic loading test for endosseous dental implants), the general principles for fatigue testing state that “at least two, and preferably three, specimens shall be tested at each of at least four loads. Moreover, “at least three specimens shall be tested, and every specimen shall reach the specified number of cycles with no failures” in order to reach the infinite life range. For all these reasons, a minimum of 9 specimens are necessary (in our study there were 10 samples in the experimental group and 9 in the control group) to meet the requirements of the International Standard.
Nevertheless, like the reviewer, the authors are aware that the sample size could not be determined a priori and that the number of specimens, although it meets the requirements of the International Standard, is limited. These facts are due to the previously mentioned requirements of the International Standard. Thus, it seems prudent to avoid establishing causal relationships or generalizing the results, at least until new studies corroborate or refute the observed findings. Accordingly, the statistical analysis derived from the fatigue tests is eminently descriptive. Likewise, it was also not possible to make a post-hoc power analysis because the 3 consecutive samples without failure that determined the infinite life range did so with the same load (i.e., standard deviation = 0).
- No information or details provided about type of the implant, characteristics, surface
More details of the implant have been introduced in Materials and methods.
- The distribution of samples in both groups is not clear especially procedures related to the control group.
The authors have clarified this comment according to the reviewer.
- Was the implant surfaces roughened by the operators or were originally rough and how the level of roughness decided.
The implants are originally rough, this aspect have been explained in the text. The authors have introduced the method sand blasting by alumina and the values of Ra.
- The IP procedure was carried out before mounting the implants in the resin cast. This is far from what is done clinically or at least after mounting. Although some steps were taken to avoid aggressive threats removal but the direct access and close manual manipulation does not resemble clinical settings. This is a major flaw in the design
-No appropriate citations were made for any IP procedure nor for roughness, fatigue and corrosion
The authors have changed 6 references according to the comment of reviewer.
- Figure 2 legends are not adequate with some text missing. Burs mixed up with implants.
This legend has been improved.
- Results
- It is not clear in the figure of fractography what regions or levels appear
A new paragraph has been introduced to explain with more detail the fracture. The figure legend has been clarified.
- Study limitations should be added at the end.
Done
Language editing is needed, many typos.
The text has been revised
Reviewer 3 Report
Comments on Font et al:
This manuscript shows rich content, providing a deep insight for some works: the study is within the journal’s scope, and I found it to be well-written, providing sufficient information. Even if the manuscript provides an organic overview, with a densely organized structure and based on well-synthetized evidence, there are some suggestions necessary to make the article complete and fully readable. For these reasons, the manuscript requires major changes.
Please find below an enumerated list of comments on my review of the manuscript:
INTRODUCTION:
LINE 26: a full stop is needed after “resistance”.
LINE 69: there is a typing error (lossanf). Please, correct this sentence.
This study demonstrates the decrease in the corrosion resistance at the interface smooth-roughened dental implant; it would be interesting to explore this aspect in relation to daily oral hygiene products. To investigate this topic, some papers about the corrosion and the residual on the titanium surfaces of dental implants, linked to daily oral hygiene products, should be mentioned. Among the many, we suggest the following one: Bianchi S, Fantozzi G, Bernardi S, Antonouli S, Continenza MA, Macchiarelli G. Commercial oral hygiene products and implant collar surfaces: Scanning electron microscopy observations. Can J Dent Hyg. 2020 Feb 1;54(1):26-31. PMID: 33240361; PMCID: PMC7533810
The main topic is interesting, and certainly of great clinical impact. As regards the originality and strengths of this manuscript, this is a significant contribute to the ongoing research on this topic.
There is a specific and detailed explanation for the methods used in this study: this is particularly significant, since the manuscript relies on a multitude of recent evidence, to derive its conclusions.
The conclusion of this manuscript is perfectly in line with the main purpose of the paper: the authors have designed and conducted the study properly. As regards the conclusions, they are well written and present an adequate balance between the description of previous findings and the results presented by the authors.
Finally, this manuscript also shows a basic structure, properly divided and looks like very informative on this topic. Furthermore, figures and tables are complete, organized in an organic manner and easy to read.
In conclusion, this manuscript is densely presented and well organized, based on well-synthetized evidence. The authors were lucid in their style of writing, making it easy to read and understand the message, portrayed in the manuscript. Besides, the methodology design was appropriately implemented within the study. However, many of the topics are very concisely covered. This manuscript provided a comprehensive analysis of current knowledge in this field. Moreover, this research has futuristic importance and could be potential for future research. However, major concerns of this manuscript are with the introductive section: for these reasons, I have major comments for this section, for improvement before acceptance for publication. The article is accurate and provides relevant information on the topic and I have some major points to make, that may help to improve the quality of the current manuscript and maximize its scientific impact. I would accept this manuscript if the comments are addressed properly.
Author Response
REVIEWER 3
Dear Reviewer,
Thanks for taking the time to review our manuscript and suggest to us to improve our work by providing a lot more detail. We have done so, and we are now submitting a manuscript that not only addresses the points you specifically raised but also many others that we have considered in order to deliver what we think is a much-improved version of our work. This version includes more paragraphs, figure, English grammar revisions in all main sections, new references. Thanks a lot. We are looking forward to your comments.
Sincerely,
Francisco-Javier Gil Mur
This manuscript shows rich content, providing a deep insight for some works: the study is within the journal’s scope, and I found it to be well-written, providing sufficient information. Even if the manuscript provides an organic overview, with a densely organized structure and based on well-synthetized evidence, there are some suggestions necessary to make the article complete and fully readable. For these reasons, the manuscript requires major changes.
Please find below an enumerated list of comments on my review of the manuscript:
INTRODUCTION:
LINE 26: a full stop is needed after “resistance”.
Done
LINE 69: there is a typing error (lossanf). Please, correct this sentence.
Corrected
This study demonstrates the decrease in the corrosion resistance at the interface smooth-roughened dental implant; it would be interesting to explore this aspect in relation to daily oral hygiene products. To investigate this topic, some papers about the corrosion and the residual on the titanium surfaces of dental implants, linked to daily oral hygiene products, should be mentioned. Among the many, we suggest the following one: Bianchi S, Fantozzi G, Bernardi S, Antonouli S, Continenza MA, Macchiarelli G. Commercial oral hygiene products and implant collar surfaces: Scanning electron microscopy observations. Can J Dent Hyg. 2020 Feb 1;54(1):26-31. PMID: 33240361; PMCID: PMC7533810
This suggestion has been incorporated in the discussion and the new reference has been also introduced.
The main topic is interesting, and certainly of great clinical impact. As regards the originality and strengths of this manuscript, this is a significant contribute to the ongoing research on this topic.
There is a specific and detailed explanation for the methods used in this study: this is particularly significant, since the manuscript relies on a multitude of recent evidence, to derive its conclusions.
The conclusion of this manuscript is perfectly in line with the main purpose of the paper: the authors have designed and conducted the study properly. As regards the conclusions, they are well written and present an adequate balance between the description of previous findings and the results presented by the authors.
Finally, this manuscript also shows a basic structure, properly divided and looks like very informative on this topic. Furthermore, figures and tables are complete, organized in an organic manner and easy to read.
In conclusion, this manuscript is densely presented and well organized, based on well-synthetized evidence. The authors were lucid in their style of writing, making it easy to read and understand the message, portrayed in the manuscript. Besides, the methodology design was appropriately implemented within the study. However, many of the topics are very concisely covered. This manuscript provided a comprehensive analysis of current knowledge in this field. Moreover, this research has futuristic importance and could be potential for future research. However, major concerns of this manuscript are with the introductive section: for these reasons, I have major comments for this section, for improvement before acceptance for publication. The article is accurate and provides relevant information on the topic and I have some major points to make, that may help to improve the quality of the current manuscript and maximize its scientific impact. I would accept this manuscript if the comments are addressed properly.
Thank you very much for your comments. The authors have improved the introduction according to the reviewer.
Reviewer 4 Report
Dear authors. Thank you for submitting the manuscript “Effect of implantoplasty on roughness, fatigue and corrosion behavior of narrow diameter dental implants”. Here is my review:
There is an excessive self-citation from author Javier Gil and it is considered unethical and falls under the subset of citation manipulation, so you must remove/replace all his references.
In the introduction, please start defining peri-implant mucositis and peri-implantitis before you mention them as a common complication and their differences.
Also please define and/or give examples of your statement about “Nonsurgical treatment has been shown of offer limited efficacy in the remission of peri-implantitis”.
In the resin cast preparation, please mention include an image for it.
In the discussion section please include more rationale for providing 5,000,000 cycles such as previous similar studies with same number of cycles (you gave some data in the fatigue testing but provide it in more detail).
Please provide power analysis (significant level, statistical power and effect size) for only using 20 tapered and 10 with hexagonal connection samples.
Thank you.
Author Response
REVIEWER 4
Dear Reviewer,
Thanks for taking the time to review our manuscript and suggest to us to improve our work by providing a lot more detail. We have done so, and we are now submitting a manuscript that not only addresses the points you specifically raised but also many others that we have considered in order to deliver what we think is a much-improved version of our work. This version includes more paragraphs, figure, English grammar revisions in all main sections, new references. Thanks a lot. We are looking forward to your comments.
Sincerely,
Francisco-Javier Gil Mur
Dear authors. Thank you for submitting the manuscript “Effect of implantoplasty on roughness, fatigue and corrosion behavior of narrow diameter dental implants”. Here is my review:
There is an excessive self-citation from author Javier Gil and it is considered unethical and falls under the subset of citation manipulation, so you must remove/replace all his references.
Yes. Four references have been changed for others in order to reduce the self-citation.
In the introduction, please start defining peri-implant mucositis and peri-implantitis before you mention them as a common complication and their differences.
Done
Also please define and/or give examples of your statement about “Nonsurgical treatment has been shown of offer limited efficacy in the remission of peri-implantitis”.
A new paragraph has been introduced in the text according to the reviewer
In the resin cast preparation, please mention include an image for it.
A new figure and explanation have been introduced.
In the discussion section please include more rationale for providing 5,000,000 cycles such as previous similar studies with same number of cycles (you gave some data in the fatigue testing but provide it in more detail). Please provide power analysis (significant level, statistical power and effect size) for only using 20 tapered and 10 with hexagonal connection samples.
According to European Standard EN ISO 14801:2016 (Dentistry – Implants – Dynamic loading test for endosseous dental implants), the general principles for fatigue testing state that “at least two, and preferably three, specimens shall be tested at each of at least four loads. Moreover, “at least three specimens shall be tested and every specimen shall reach the specified number of cycles with no failures” in order to reach the infinite life range. For all these reasons, a minimum of 9 specimens are necessary (in our study there were 10 samples in the experimental group and 9 in the control group) to meet the requirements of the International Standard.
Nevertheless, like the reviewer, the authors are aware that the sample size could not be determined a priori and that the number of specimens, although it meets the requirements of the International Standard, is limited. These facts are due to the previously mentioned requirements of the International Standard. Thus, it seems prudent to avoid establishing causal relationships or generalizing the results, at least until new studies corroborate or refute the observed findings. Accordingly, the statistical analysis derived from the fatigue tests is eminently descriptive. Likewise, it was also not possible to make a post-hoc power analysis because the 3 consecutive samples without failure that determined the infinite life range did so with the same load (i.e., standard deviation = 0).
Round 2
Reviewer 3 Report
Authors complied to the suggestions.
Manuscript can be accepted